



# Effect of aerosol composition on the performance of low-cost optical particle counter correction factors

Leigh R. Crilley[1,#], Ajit Singh[1], Louisa J. Kramer[1], Marvin D. Shaw[2], Mohammed S. Alam[1],
Joshua S. Apte[3], William J. Bloss[1], Lea Hildebrandt Ruiz[3], Pingqing Fu[4,5], Weiqi Fu[5],
Shahzad Gani[3], Michael Gatari[6], Evgenia Ilyinskaya[7], Alastair C. Lewis[2], David Ng'ang'a[6],
Yele Sun[5], Rachel C. W. Whitty[7], Siyao Yue[5], Stuart Young[2] and Francis D. Pope[1]

[1]School of Geography, Earth and Environmental Sciences, University of Birmingham, Birmingham, UK

[2]National Centre for Atmospheric Science, Wolfson Atmospheric Chemistry Laboratories, University of York, York, UK

[3]Department of Civil, Architectural and Environmental Engineering, The University of Texas at Austin, Austin, Texas, USA

[4]Institute of Surface-Earth System Science, Tianjin University, Tianjin, China

[5]Institute of Atmospheric Physics, Chinese Academy of Sciences, Beijing, China

[6]Institute of Nuclear Science and Technology, University of Nairobi, Nairobi, Kenya

[7]School of Earth and Environment, University of Leeds, Leeds, UK

#now at: Department of Chemistry, York University, Toronto, Canada.

*Correspondence to*:

Leigh R. Crilley: lcrilley@yorku.ca

**Abstract.** There is considerable interest in using low-cost optical particle counters (OPC) to supplement existing routine air quality networks that monitor particle mass concentrations. In order to do this, low-cost OPC data needs to be cross-comparable with particle mass reference instrumentation, and as yet, there is no widely agreed methodology. Aerosol hygroscopicity is known to be a key parameter to consider when correcting particle mass concentrations derived from a low-cost OPC, particularly at high ambient Relative Humidity (RH). Correction factors have been developed that apply κ-Köhler theory to correct for the influence of water uptake by hygroscopic aerosols. We have used datasets of co-located reference particle measurements and a low-cost OPC (OPC-N2, Alphasense), collected in four cities in three continents, to explore the performance of this correction factor. We report evidence that the elevated particle mass concentrations, reported by the low-cost OPC relative to reference instrumentation, is due to bulk aerosol hygroscopicity under different RH conditions, which is determined by aerosol composition and in particular the levels of hygroscopic aerosols (sulphate and nitrate). We exploit measurements made in volcanic plumes in Nicaragua, that are predominantly composed of sulphate aerosol, as a natural experiment to demonstrate this behaviour in the ambient atmosphere, with the observed humidogram closely resembling the calculated pure sulphuric acid humidogram. The results indicate that the particle mass concentrations derived from low-cost OPCs during periods of high RH (> 60 %) need to be corrected for aerosol hygroscopic growth. We employed a correction factor based on κ-Köhler theory and observed corrected OPC-N2 PM$_{2.5}$ mass concentrations to be within 33% of reference measurements at all sites. The results indicated that an *in situ* derived κ (using suitable reference instrumentation) would lead to the most accurate correction relative to co-located reference instruments. Applying literature κ in the correction factor also resulted in improved performance of OPC-N2, to be within 50 % of reference. Therefore, for areas where suitable reference



instrumentation for developing a local correction factor is lacking, using a literature κ value can result in a reasonable correction. For locations with low levels of hygroscopic aerosols and RH, a simple calibration against gravimetric measurements (using suitable reference instrumentation) would likely be sufficient. Whilst this study generated correction factors specific for the Alphasense OPC-N2 sensor, the calibration methodology developed
is likely amenable to other low cost PM sensors.

### 1.0   Introduction

Advances in miniaturization technology have led to the development of many different kinds of low-cost air pollution sensors, ranging from passive gas samplers to miniaturized versions of reference instruments (Lewis et al., 2018;Jayaratne et al., 2018). The term low-cost is relative and typically refers to the sensor being at least an
order of magnitude cheaper than corresponding reference instrumentation (Lewis et al., 2018). Monitoring of key air pollutants (e.g. $PM_{2.5}$, $NO_x$ and $O_3$) has traditionally been performed via reference standard or equivalent monitors at fixed monitoring stations. However, this approach can lack the necessary spatial coverage to properly assess personal exposure due to significant spatial heterogeneity in the concentration of air pollutants in urban areas (de Nazelle et al., 2017). Low-cost sensors are an attractive option due to their reduced costs and portability,
making them viable for mobile or highly spatially resolved measurements, to complement existing monitoring frameworks. This has led to low-cost sensors becoming a common feature of an increasing number of air pollution monitoring operations (Snyder et al., 2013;Morawska et al., 2018).

The trade-off with using low-cost sensors are that they are currently not as accurate, precise, selective or sensitive when compared to research or regulatory grade instrumentation (Mead et al., 2013;Lewis et al., 2018;Lewis et al.,
2016;Smith et al., 2017;Crilley et al., 2018;Borrego et al., 2016;Popoola et al., 2016). Consequently, low-cost sensors of air pollutants need to be carefully characterised to ensure they meet the specific requirements of the intended application (Castell et al., 2017). In their review, Morawska et al. (2018) concluded that low-cost sensors were fit for purpose for many applications, such as supplementing routine air quality measurements and engaging the public and community groups. However, there is still work needed if low-cost sensors are to be used for
accurate exposure measurements, or in the future for compliance monitoring, which is of particular interest in under-monitored low and middle-income countries (LMICs). LMICs typically have high urban air pollution yet the resources and infrastructure are sometimes lacking to support continuous classical reference air quality measurements (Pope et al., 2018). One of the challenges with using low-cost sensors in this setting is there is currently no agreed methodology for the evaluation of their accuracy and precision and their subsequent
calibration (Lewis et al., 2018).

A key pollutant for air monitoring networks is airborne particulate matter (PM) due to their well-established detrimental physical health effects (Cohen et al., 2005;Landrigan et al., 2018). In particular, exposure to fine particles ($PM_{2.5}$, particles with an aerodynamic diameter less than 2.5 µm) is known to have multiple disease pathways (Landrigan et al., 2018). Recently, short term exposure to $PM_{2.5}$ has been linked to short term cognitive
decline (Shehab and Pope, 2019). $PM_{2.5}$ mass regulatory limits are based on the dry particle mass concentration and regulatory-grade particle mass instrumentation dry the aerosol before measurement to record the dry aerosol mass concentrations. Low-cost optical particle counters (OPC) measure particle diameter and number concentrations by light scattering and convert this to particle mass concentrations by assuming particle sphericity and a uniform density. Low-cost OPCs typically do not dry the aerosol before measurement, and this can result





in over-estimation of the dry particle size (compared to that which would be determined after drying) under high RH conditions (Crilley et al., 2018;Jayaratne et al., 2018;Di Antonio et al., 2018), thought to be related to the uptake of water by hygroscopic aerosol. Consequently, the reported PM mass concentrations by low-cost OPC are of the wet particle mass concentration and these need to be converted to a dry particle mass concentration in order to have comparability with regulatory standards and reference instrumentation. One solution to measuring the dry mass of particles would be to add a pre-conditioning drying step prior to the OPC, but this would result in higher costs, greater power consumption and less instrument portability thereby reducing the unique selling points (USPs) of low-cost sensor devices.

Recently, a methodology to correct the wet particle mass concentrations to dry mass concentrations was proposed by Crilley et al. (2018) based upon the κ-Köhler theory (Petters and Kreidenweis, 2007). κ-Köhler theory describes the relationship between particle hygroscopicity and volume by a single value, κ, and can be adapted to relate particle mass to average bulk hygroscopicity at a given RH. Crilley et al. (2018) calculated κ values representative of ambient bulk hygroscopicity using co-located reference instruments to derive a correction factor for the derived OPC-N2 (Alphasense) particle mass concentrations. Application of this *in situ* correction factor by Crilley and co-workers notably improved the OPC-N2 reported $PM_{2.5}$ and $PM_{10}$ mass concentrations, to be within 33 % of the reference instrumentation at urban background location with high ambient RH. Subsequently, Di Antonio et al. (2018) proposed a similar method that applied κ-Köhler theory to correct the particle size distribution measured by the OPC-N2, and then calculated the particle mass fraction concentration using this corrected particle size. Using this approach Di Antonio and co-workers also observed notable improvement in the OPC-N2, to be within 43 % of reference $PM_{2.5}$ mass concentrations. Di Antonio and co-workers assumed κ values for their correction factor, based upon the assumed major hygroscopic components of the aerosol mix (ammonium sulphate and sodium chloride), which may not be realistic considering the complex multi-compositional nature of urban particles.

It is clear that the aerosol hygroscopicity is a key parameter to consider when correcting particle mass concentration derived by a low-cost OPC (Crilley et al., 2018;Di Antonio et al., 2018). Aerosol hygroscopicity is dependent on the chemical compounds present, and consequently the derived correction factor may vary from location to location due to differences in particle bulk composition, shape and density. To investigate this, we utilised datasets containing co-located particle measurements from reference instruments and a low-cost OPC collected in four cities in three continents: Birmingham, UK; Nairobi, Kenya; Delhi, India and Beijing, China. Across these four cities, the airborne particle composition and range of ambient RH varied considerably allowing for the exploration of sensor performance in response to these factors (composition and RH) and how it affected the calculated correction factors. We also report measurements taken near a volcano in Nicaragua, at a location that received regular volcanic plumes that contained particles that were typically chemically homogenous.

### 2.0 Method

The datasets used in the current work were acquired in several different field campaigns, at different times, but the same type of low-cost particle sensor was deployed at all five locations, the OPC-N2 manufactured by Alphasense. This sensor has been described in detail in Sousan et al. (2016) and Crilley et al. (2018) and can be considered as a miniaturized optical particle counter. The measured particle number concentration by the OPC-N2 is converted via on board factory calibration to particle mass concentrations for $PM_1$, $PM_{2.5}$ and $PM_{10}$ size



fraction according to European Standard EN481 (OPC-N2 manual). Data collection with the OPC-N2 followed the procedures outlined in Crilley et al. (2018) at the four urban locations via a Raspberry Pi employing the py-opc Python library (Hagan et al., 2018). Uncorrected PM mass concentrations were used without any modification. In Birmingham, Delhi, Beijing and Nairobi the same inlet length (12 cm 3/8" dia. stainless steel tubing) was used

for each OPC-N2. At Nicaragua, the OPC-N2 was part of the commercially available AQMesh system (Air Monitors), with uncorrected PM concentrations extracted.

### 2.2 Measurement Locations

Each site in this study, other than the volcanic Nicaraguan site, are classed as urban background for their respective cities. We have focused on $PM_{2.5}$ mass concentrations in this study, as this particle size fraction was measured by

reference instrumentation at all study sites. We also note that we used a different OPC-N2 sensor at each site. Previous work has shown that co-located multiple OPC-N2 have an inter-unit precision of 22±13 % for $PM_{10}$ mass concentrations (Crilley et al. 2018).

### 2.2.1 Birmingham, United Kingdom

The OPCs were deployed at two urban background locations in Birmingham. The first was the Birmingham Air

Quality Supersite at Elms Road  (BAQS, 52.4554 N, 1.9286 W), located with the University of Birmingham campus (Alam et al., 2015) and will be referred to as Bham BAQS throughout. The second site was the Tyburn Road air monitoring station, part of the UK Automatic Rural Urban Network (AURN), referred to as Bham Tyburn throughout. This dataset has been previously described in Crilley et al. (2018), and the current work focuses on the long-term measurements (October 2016 till February 2017) at Bham BAQS, using the OPC-N2 that gave the

most complete time series. The reference instrument for $PM_{2.5}$ mass concentrations measurements at Bham Tyburn was a Tapered Element Oscillating Microbalance with a Filter Dynamic Measurement System (TEOM-FDMS). The Bham Tyburn dataset is used to compare to other sites that had the same reference instrument (Beijing and Delhi, See Sections 2.2.2 and 2.2.3). The reference instrument at Bham BAQS was a GRIMM Portable Aerosol Sampler (model 1.108) that was serviced and calibrated before the measurements. The GRIMM is an OPC type

device similar to the low cost sensors, but does contain a pre-conditioning step that reduces the internal RH of the device.  Previous work demonstrated that the GRIMM was not affected by RH, based on co-located measurements with a TEOM-FDMS (Crilley et al., 2018).

### 2.2.2 Beijing, China

The measurements in Beijing formed part of the Air Pollution and Human Health in a Chinese mega-city (APHH-

Beijing, www.aphh.org.uk), a joint UK-China program addressing air quality in Beijing (Shi et al., 2019). The measurements took place at the Chinese Academy of Science Institute of Atmospheric Physics (IAP) tower campus (39.9735 N, 116.3723 E), located in the northern suburbs of Beijing. The OPC-N2 sampled on top of a shipping container, with a height of approx. 2.5 m from 5th-9th December 2016. In addition, co-located ground level measurements with a TEOM-FDMS set to measure $PM_{2.5}$ mass concentrations and an Aerodyne Aerosol

Mass Spectrometer (AMS) were obtained (Xu et al., 2019).

### 2.2.3 Delhi, India

The measurements in Delhi were part of the Air Pollution and Human Health in an Indian mega-city (APHH-Delhi, www.urbanair-india.org), a joint UK-India project tackling air pollution in Delhi. The sampling location



was Indian Institute of Technology Delhi (IITD) main campus in Hauz Khas (28.5464 N, 77.1913 E), located in the southern suburbs of New Delhi. The instruments were located on the roof (4 stories) of Block IV at IITD. The inlet for the co-located $PM_{2.5}$ TEOM-FDMS was approximately 5 m from the OPC-N2 at the same sampling height. On-line measurements of inorganic aerosol concentrations were provided by an Aerodyne Aerosol Chemical Speciation Monitor (ACSM), located nearby on the IITD campus in Block V, at the same sampling height (Gani et al., 2019).

**2.2.4 Nairobi, Kenya**

The measurements in Nairobi are previously reported in Pope et al. (2018). These measurements are part of the 'A Systems Approach to Air Pollution' program (ASAP East Africa, www.asap-eastafrica.com). In the current work, we used the urban background data that were collected on the rooftop of the American Wing building at the University of Nairobi (1.2801 S, 36.8163 E). The sampling inlet was at a height of 17 m above ground level with unobstructed airflow in all directions. The measurement period is from 2 February to 24 March 2017. Calibration of the OPC was carried out in situ using a standardized gravimetric approach by collocation with of the OPC and an Anderson dichotomous impactor (Sierra Instruments Inc., USA).

**2.2.5 Masaya volcano, Nicaragua**

Masaya is an active volcano that is currently degassing and due to its low altitude (600 m a.s.l.) the volcanic plume causes persistent gas and PM air pollution in nearby populated areas. The presented results are part of the first study of high temporal and long-term measurements of PM and $SO_2$ concentrations in several populated areas near Masaya volcano. Here we will discuss the results from station 789, set up in Pacaya community (11.9553°N, 86.3013°W, 870 m a.s.l.) 15 km to the west of Masaya volcano. Because it is located at a higher altitude than the degassing crater, the volcanic plume frequently grounds there. The station was set up at Susie Syke private clinic, on a post approximately 6 m above ground level, where it was not obstructed by vegetation, buildings or other objects. The site is not believed to be influenced by firewood burning. It is located ~100 m from a paved highway (busy during morning and evening rush hours) but is upwind of it during the predominant weather conditions.

Measurements were performed using an 'AQMesh' pod, a commercially available sensor package. $SO_2$ concentrations were measured by an Alphasense B4-series electrochemical sensor; while particle concentrations were measured by the OPC-N2. Note uncorrected particle mass concentrations were extracted from the AQMesh. The AQMesh was operational between 27 February 2017 and 15 December 2017. Gaps in the data time series are due to power outages.

**2.3 Description of applied correction factor**

The methodology for the applied correction factor has previously been described in detail by Crilley et al. (2018). Briefly, the correction factor uses κ-Köhler theory to relate the particle mass to hygroscopicity for a given RH (Pope, 2010), according to Eq (1):

$$a_w = \frac{(m/m_o - 1)}{(m/m_o - 1) + (\frac{p_w}{\rho_p}\kappa)} \tag{1}$$

Where $a_w$ is the water activity ($a_w$ = RH/100), $m$ and $m_o$ are the wet and dry (RH=0 %) particle mass, respectively while $p_w$ and $p_p$ are the density of the dry particles and water, respectively. The value for κ, which relates the bulk





aerosol composition to hygroscopicty, can be determined by a non-linear curve fitting of a humidogram, calculated using the ratio of a wet/dry particle mass as a function of water activity ($a_w$, RH/100). Throughout, we used the raw mass concentrations as reported by the OPC-N2. We used the reference instrument measurements (TEOM-FDMS and GRIMM, as indicated in Section 2.2) as the dry particle mass while the raw OPC-N2 is the wet particle

mass concentration. The TEOM-FDMS employs a Nafion dryer and therefore measures dry particle mass concentration (Grover et al., 2006). Equation (1) can be rearranged to calculate the correction factor $C$ according to Eq. (2):

$$C = 1 + \frac{\frac{\kappa}{\rho_p}}{-1 + \frac{1}{a_w}} \qquad (2)$$

The OPC-N2 assumes the ambient particle density to be 1.65 g cm$^{-3}$ across all size bins to derive the particle mass
concentrations from the measured particle number concentrations (Crilley et al., 2018), therefore we have used this density for the dry particles ($\rho_p$) in Eq. (2). We also note that we assume both the OPC-N2 and reference instrument responses are linear over the range of measured concentrations at each site. The raw particle mass concentration derived by the OPC-N2 are corrected according to Eq. (3):

$$PM_{corr} = \frac{PM_{raw}}{C} \qquad (3)$$

**3 Results and Discussion**

A wide range of ambient particle concentrations and relative humidity was observed across the different measurement locations, with an overview provided in Table 1. Typically, low particle concentrations were observed in Birmingham and Nairobi with higher levels of humidity in Birmingham compared to Nairobi (Table
1). Meanwhile a high particle load was observed in Delhi and Beijing, as would be expected for winter in these two cities. In Delhi, there was a wide range of humidities observed (10-100 %), while in Beijing it was relatively dry (mean of 36±15 %) during the measurement periods. The observed difference in particle load and composition between sites allows for the effect of relative humidity on the OPC-N2 measurements and the applied correction factor to be explored in the following sections.

**Table 1:** Summary of measurement datasets. Reported OPC-N2 PM$_{2.5}$ mass concentrations are uncorrected. For the Nicaragua measurements there was no co-located reference instrumentation. Only one 24 hr average gravimetric PM$_{2.5}$ concentration was available for Nairobi, presented with stated measurement uncertainty.

| Site | Date | RH (mean ± s.d) | OPC-N2 PM$_{2.5}$ (µg m$^{-3}$) | Reference PM$_{2.5}$ (µg m$^{-3}$) |
|---|---|---|---|---|
| **Birmingham** | Oct 2016- Feb 2017 | 89±10 % | 0.3-566 | 0.5-63 |
| **Beijing** | Dec 2016 | 36±15 % | 3-274 | 2.7-208 |
| **Delhi** | Jan-Feb 2018 | 59±25 % | 12-1113 | 50-478 |
| **Nairobi** | Feb-Mar 2017 | 51±18 % | 4-135 | 27.6±6.8 |
| **Nicaragua** | Feb-Dec 2017 | 77±11 % | 0.5-742 | NA |

**3.1 Effect of RH on measurement by OPC-N2 at all sites**

To explore if there was evidence for an artefact on the OPC-N2 derived PM$_{2.5}$ mass concentrations due to RH, we plotted the reported PM$_{2.5}$ mass concentrations by the OPC-N2 as a function of RH (Fig 1). From Fig 1, there is clear influence of RH on the measurements performed in Delhi and Birmingham, evidenced by the observed exponential increase in particle mass with RH (Figs 1a and 1d). At Beijing, the observed stepwise increase in the



derived measured particle mass between a RH of 40-50% RH may point to deliquescence of a predominant PM component (Fig 1c), explored further in later sections. What is evident from these three sites (Beijing, Birmingham, Delhi) was there was a large spread in derived $PM_{2.5}$ mass concentrations at high RH, which likely reflects the heterogeneous nature of the particle composition and hence hygroscopicity.

5    Meanwhile at Nairobi, the derived concentrations by the OPC-N2 appeared to be independent of RH. Typically during the dry season in Nairobi, airborne mineral dust comprises a large fraction of $PM_{2.5}$ (35% annual mean, (Gaita et al., 2014), which is known to have low hygroscopocity. Furthermore, we note that the measurements in Nairobi were performed during the dry season and as a result the ambient RH was typically less than 85%, the RH value identified in Crilley et al. (2018) where the OPC-N2 becomes significantly sensitive to RH. This

10   combined with the low hygroscopocity of the aerosol in Nairobi, was the likely reason why there was little evidence for an RH artefact observed for the OPC-N2 (Pope et al., 2018). Therefore, a simple calibration against gravimetric measurements is likely to be sufficient in locations with low RH and low proportion of hygroscopic aerosols, such as Nairobi.

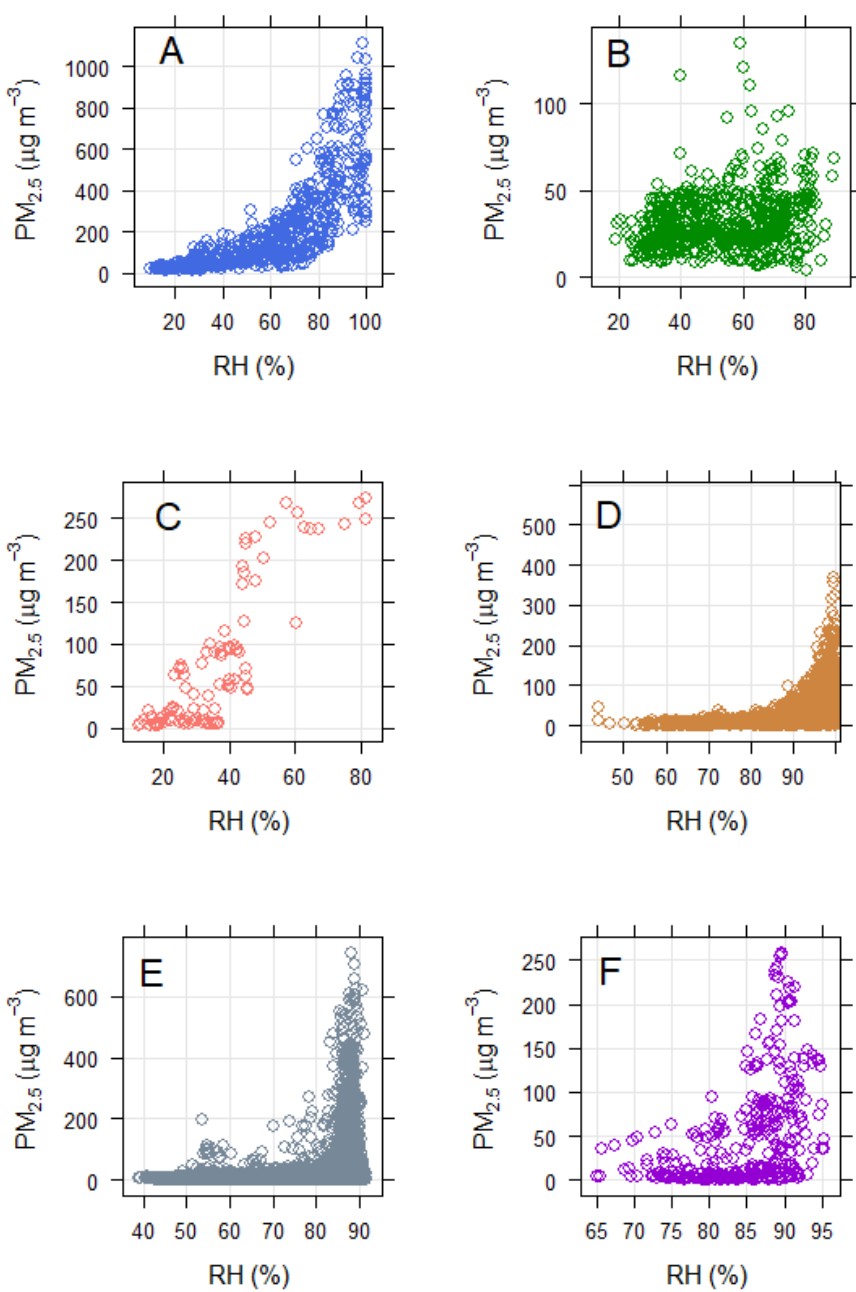

**Figure 1:** Plot of reported PM$_{2.5}$ mass concentration by OPC-N2 against ambient RH for the whole measurement
period in Delhi (A), Nairobi (B), Beijing (C), Bham BAQS (D), Nicaragua (E), Bham Tyburn (F). Note the
different y- and x-axis scales.



As posited in Crilley et al. (2018), the RH artefact on the OPC-N2 was likely related to the ambient aerosol bulk hygroscopicity. Therefore, we plotted humidograms for sites from Fig. 1 with evidence for an RH effect where reference particle mass concentration data was available (i.e. Birmingham, Delhi and Beijing; Table 1), shown in Fig 2. In Beijing, there was insufficient data at high RH due to the short time period of sampling (4 days), and the

5    factors affecting the response of the OPC-N2 explored in more detail in Section 3.3. A quasi-exponential increase in the ratio of OPC-N2 to reference instrument concentrations at high RH was observed at Birmingham and Delhi, as would be expected if the aerosols were undergoing hygroscopic growth (Fig. 2). Using κ-Köhler theory (Petters and Kreidenweis, 2007), calculated κ were 0.1 and 0.16 for Birmingham and Delhi, respectively. These κ values are typical for continental regions with high organic loadings (Pringle et al., 2010). While high organic loadings

10    would be expected for Delhi during winter, there is also significant loadings of hygroscopic aerosols such as sulphate and nitrate (Gani et al., 2019) and this is explored in more detail later. Differences in aerosol composition would likely explain why the calculated κ at Bham BAQS (0.1) was lower than that observed at Bham Tyburn (0.38-0.41, Crilley et al. 2018). Previous work in Birmingham demonstrated that the proportion of ammonium sulphate and nitrate decreases in winter compared to summer (Yin et al., 2010), and may explain the observed

15    lower κ value over winter. What becomes evident from Figure 2 is that different κ values were observed at each site. If the aerosol composition was broadly similar at each site, we would expect the same κ value. This suggests that the aerosol composition varies significantly over the different measurement sites.

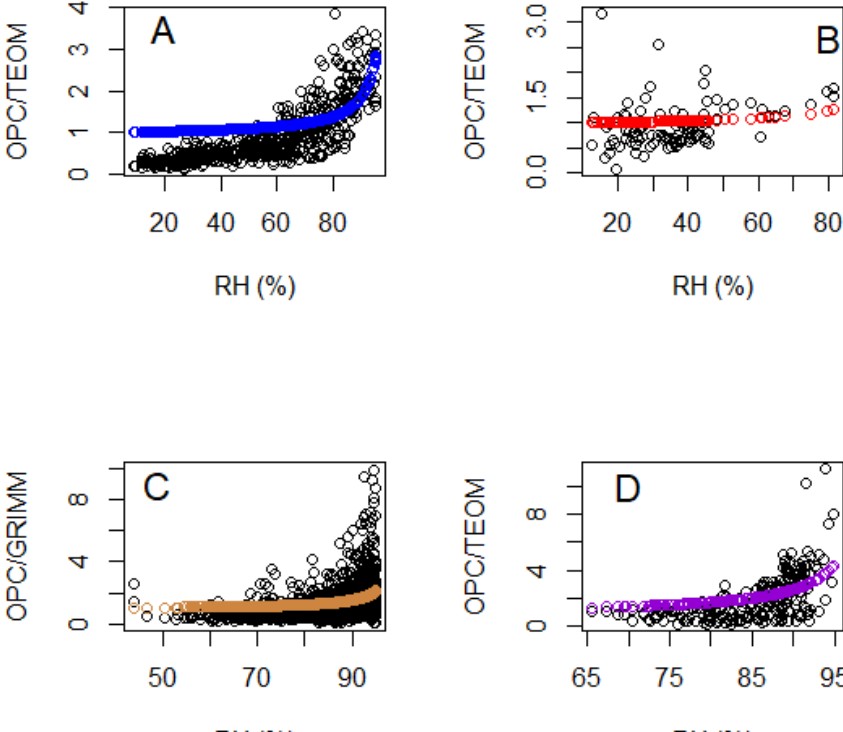

**Figure 2:** Humidograms with corresponding κ fit for Delhi (A), Beijing (B), Bham BAQS (C), Bham Tyburn
(D). Note the different y-axis and x-axis scales. The two-stage correction factor described in Section 3.3.1 has
not been applied for these humidograms.

### 3.2 Effect of aerosol composition on OPC-N2 RH correction factor

### 3.2.1 Mixed aerosol composition (Urban)

To explore the effect of aerosol composition on correction factor for the OPC-N2, we focus first on the Beijing
and Delhi measurements, as co-located on-line aerosol composition data was available at these two sites. During
the measurements in Beijing, there were periods when the OPC-N2 and TEOM were in reasonable agreement,
typically at lower PM concentrations as observed by the regulatory grade equipment. Fig. 3a demonstrates that at
concentrations below ca. 150 µg m$^{-3}$, there was linear relationship between reported OPC-N2 and the TEOM
concentrations ($r^2$ of 0.85), with a slope of 0.72. When the PM$_{2.5}$ mass concentrations were above 150 µg m$^{-3}$, the
relationship appeared to deviate from linearity, but these were also the times when the RH was higher (>50 %,
Fig. 3a). Generally, the times of high RH also corresponded to times of relatively high sulphate concentrations
(Fig. 3b) and to a lesser extent high particle nitrate concentration (Fig. S1, Supporting Information). Both nitrate
and sulphate aerosol have high hygroscopicities (Petters and Kreidenweis, 2007).



Similar trends were also observed in the Delhi measurements. Generally, at low RH there appears to be a linear relationship between the reported OPC-N2 and TEOM concentrations (Fig. 4a), but this deviates from linearity at high RH, similar to that observed in Beijing. For periods where the ambient RH in Delhi was less than 50 %, we observed that the OPC-N2 generally recorded $PM_{2.5}$ mass concentrations half that of the TEOM (slope of 0.48

and $r^2$ of 0.55). This was broadly similar to that observed in Beijing (Fig. 3a) and suggests that the OPC-N2 generally under-reports $PM_{2.5}$ mass concentrations at low RH (<50 %). Also apparent from Fig. 4a, there were times in Delhi when the RH was high (>80 %) and yet the $PM_{2.5}$ mass concentrations by the OPC-N2 were of a similar relationship to the TEOM at low RH.

We therefore plotted the relationship of OPC-N2 and TEOM $PM_{2.5}$ mass concentrations coloured by the sum of

sulphate and nitrate concentrations (Fig. 4b) and generally, when the concentration of these species and RH were high, we observed notably higher OPC-N2 concentrations relative to the reference. Both sulphate and nitrate are highly hygroscopic aerosols and this therefore suggests that the high readings by the OPC-N2 relative to the TEOM in Beijing and Delhi were due to water uptake by hygroscopic particles as suggested by Crilley et al. (2018). From Figs. 3 and 4, it appears that this effect was occurring at RH above 50 %, below the deliquescence

point of ammonium sulphate (79 %) suggesting that there was uptake by other aerosols such as nitrate (Hu et al., 2010). Nitrate aerosols also have a smoother continual take up of water (Gibson et al., 2006;Hu et al., 2010), and this points to nitrate aerosols contributing at RH below 79 %. Aerosols with multi-component mixtures are observed to deliquesce earlier than the deliquescence points of the individual components e.g. Pope et al. (2010). Overall, these results from Beijing and Delhi suggest that the applied calibration depends not just on the ambient

RH but also aerosol composition.

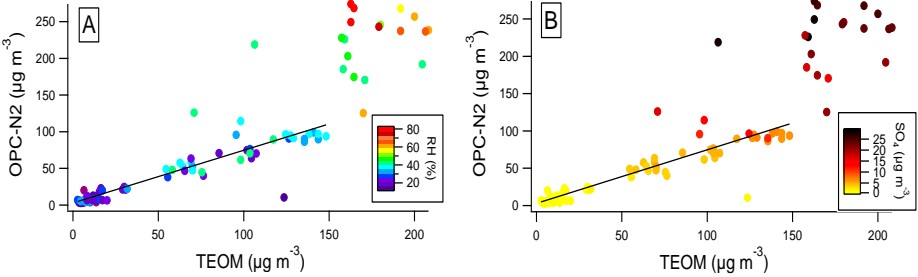

**Figure 3:** Derived OPC-N2 uncorrected $PM_{2.5}$ mass concentrations against TEOM $PM_{2.5}$ mass concentration measurements coloured by ambient RH (A) and sulphate concentration (B) in Beijing. The straight line indicates the linear regression fit for concentrations below 150 µg m$^{-3}$





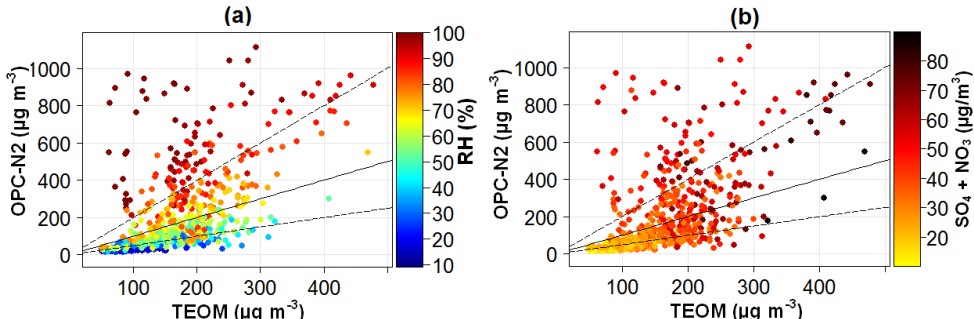

**Figure 4:** Derived OPC-N2 uncorrected PM$_{2.5}$ mass concentrations against TEOM PM$_{2.5}$ mass concentration measurements coloured by **(a)** ambient RH and **(b)** sum of particle sulphate and nitrate concentration in Delhi. The solid line is 1:1, while the dashed lines are 0.5:1 and 2:1.

### 3.2.2 Homogenous aerosol composition (volcano plume)

Fresh volcanic plumes are typically dominated in composition by sulphuric acid and therefore these plumes offer an opportunity to explore using k-Köhler theory to develop the correction factor in a substantially homogenous aerosol mix under ambient conditions. If the RH artefact is due to aerosol hygroscopicity, then the resultant humidogram using data collected by the OPC-N2 in the plume should resemble that for sulphuric acid. To derive the volcanic plume humidogram, shown in Figure 5, the following steps were taken. The plume was identified at station 789 when the collocated gas phase SO$_2$ measurement was greater than 20 ppm. The aerosol within the plume was assumed to be composed solely of sulphuric acid with a corresponding κ value of 1.19 (Wexler and Clegg, 2002). The dry mass of the volcanic particles were calculated using Eq. (1), with RH input from the collocated measurements at site 789. The derived humidogram was compared to the pure sulphuric acid humidogram calculated using E-AIM model I (Fig. 5). The observed agreement between model and measurements strongly points to particle hygroscopic growth driving the high particle mass concentrations observed by the OPC-N2 at high RH.

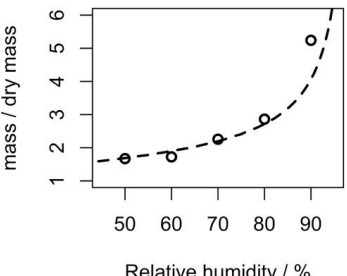

**Figure 5:** Comparison of humidograms from pure sulphuric acid and that observed in the Nicaragua volcanic plume. Circles: Nicaragua plume aerosol. Dashed line: modelled sulphuric acid humidogram from the E-AIM model.





### 3.3 Evaluation of the OPC-N2 performance in Delhi and Birmingham

During the measurements in Delhi, the OPC-N2 typically over-reported the PM$_{2.5}$ mass concentrations relative to the reference (Fig. 6A). The OPC-N2 assumes a uniform particle density of 1.65 g cm$^{-3}$ in the particle counts to mass conversion, and this density may be inappropriate for Delhi aerosol during winter. Previous measurements of aerosol density during winter in Delhi at midday were on average 1.28±0.12 g cm$^{-3}$ (Sarangi et al., 2016), lower than applied by the OPC-N2 and so this would not explain this discrepancy. From Fig. 6A, higher PM$_{2.5}$ mass concentrations relative to the reference were reported by the OPC-N2 towards the start of the measurement period (Fig. 6A), generally during the times when the RH was high (e.g. 25-29$^{th}$ Jan, Fig. 6B). We also note that the reported concentrations by the OPC-N2 appear to lose structure towards the end of the measurements in Delhi (11$^{th}$ Feb onwards, Fig. 6A), which may point to deterioration in the OPC-N2 performance (possibly due to dust coating within the instrument). This may be the reason why the OPC-N2 reported lower concentrations relative to the reference towards the end of the measurement period. As the OPC-N2 measured for 24 days in total, this suggests that the lifetime of low cost particle sensors in Delhi may only be of the order of a month.

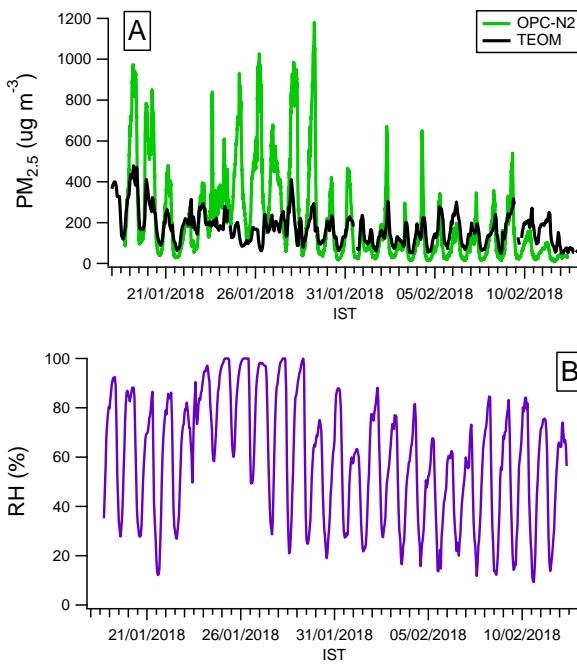

**Figure 6:** Time series of reported OPC-N2 and TEOM-FDMS PM$_{2.5}$ mass concentrations (A) and ambient RH (B) at IIT Delhi.

As the reported OPC-N2 concentrations at Birmingham and Delhi demonstrated an artefact due to RH (Fig. 2), we applied the correction factor from Crilley et al. (2018), using the *in situ* locally derived κ values. In addition, the κ for ammonium sulphate (0.61) was also used, as Di Antonio et al. (2018) suggested it may be more representative for urban aerosols. The results of the correction factors, relative to the co-located reference instruments are summarized in Table 2. Compared to the uncorrected OPC-N2 concentrations, application of correction factor with both κ resulted in improved performance of the reported concentrations relative to the





reference. However, the using the locally derived κ resulted in the best correction of the OPC-N2, to be within 33 % of the reference measurements, compared to using the ammonium sulphate κ (Table 2).

**Table 2:** Slopes of uncorrected and corrected PM$_{2.5}$ mass concentrations from the OPC-N2 relative to the reference instruments (r$^2$ in brackets). Intercepts were not constrained to zero. There were four OPC-N2 measuring at Bham Tyburn and the range is presented.

| | UNCORRECTED | CORRECTED | |
| --- | --- | --- | --- |
| | | *Locally derived κ* | *κ for NH₄SO₄* |
| **BHAM BAQS** | 3.5 (0.24) | 1.3 (0.44) | 0.5 (0.24) |
| **DELHI** | 1.73 (0.33) | 1.1 (0.60) | 0.55 (0.60) |
| **BHAM TYBURN** | 2.5-3.5 (0.64-0.67) | 0.98-1.33 (0.82-0.85) | 0.72-0.98 (0.84-0.86) |

We also observed that ratio of OPC-N2/GRIMM concentrations was low at high RH at Bham BAQS (Fig. 2C). The spread in OPC/GRIMM ratios observed at high RH at Bham BAQS is indicative of a wide range of aerosol composition with differing hygroscopocity over the 4 months. The calculated κ value for each month in Birmingham and found little variability (0.1-0.12) from October to January and suggest that on average the bulk hygroscopicity of the aerosols was consistent, though within a large range.

**3.3.1 Two-stage correction methodology for datasets with wide range of ambient RH**

For the Delhi dataset, the observed wide range of ambient RH may have affected the correction factor (Fig. 2), as particle hygroscopic growth would be limited at low RH. Consequently at low RH, defined as RH <60 %, a linear correction factor may be more appropriate. From Fig. 2, there appeared to be a linear relationship between the OPC-N2 and TEOM PM$_{2.5}$ measurements for RH less 60 %. We therefore calculated a linear correction factor for the OPC-N2 relative to the TEOM when the ambient RH was less than 60 % (Fig. S3, Supporting Information) and applied it the reported OPC-N2 PM$_{2.5}$ concentrations. Using this normalised OPC-N2 concentrations, the humidogram was replotted (Fig. 7), and the corresponding κ fit calculated (Fig. 7). Using this normalised OPC-N2 measurements, the κ line better matches the observed OPC/TEOM (Fig. 7) compared to using the uncorrected OPC-N2 data (Fig. 2). The calculated κ value from Fig. 7 was 0.45, which may be considered more realistic considering the high sulphate and nitrate loading in Delhi (Fig. 4). Using the κ from Fig. 7, we corrected the normalised OPC-N2 PM$_{2.5}$ concentrations via Eq. (2) and (3). The time series of the corrected OPC-N2 concentrations is shown in Fig. S4 (Supporting Information), and the application of this two-stage correction method resulted in the OPC-N2 being in good agreement with the reference instrument (slope of 1.1, r$^2$ of 0.61). However, we note that this is similar to the agreement observed when the OPC-N2 was corrected without the two-stage approach (Table 2). However, the two-stage approach resulted in more physically realistic humidograms and κ (Fig. 7) for Delhi and this approach may be more appropriate for locations that experience a wide range of ambient RH.

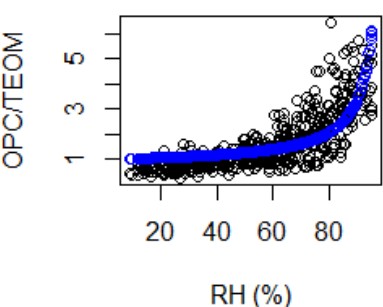

**Figure 7:** Humidograms for Delhi using normalised OPC-N2 $PM_{2.5}$ mass concentrations with corresponding κ fit.

### 3.4 On the global applicability of correction factors

The results so far point to the need to know aerosol composition in order to accurately apply a suitable correction factor, in agreement with previous work (Di Antonio et al., 2018;Crilley et al., 2018). However, to determine aerosol composition at the necessary time resolution would require expensive co-located equipment measuring aerosol composition (e.g. an Aerosol Mass Spectrometer) and somewhat negating the USP of a low cost monitor. Di Antonio et al. (2018) suggest air mass origin (via HYSPLIT) could provide compositional information in order

to determine the appropriate κ value to use in the correction. We therefore examined the long-term Birmingham dataset for times when the applied correction factor over/under-corrected the OPC-N2 mass concentrations relative to reference instrument. However, unlike Di Antonio et al. (2018) we could not find any consistent patterns with respect to air mass origin and the performance of the correction factor. Furthermore, in order to apply a correction factor via this approach would require significant post-processing time. This therefore raises

the question if it remains a 'low-cost' option.

To remain a low-cost option, a simple correction that can be applied to the OPC-N2 irrespective of aerosol composition changes is needed, though this may decrease the accuracy of the correction factor. For many locations around the world, ambient $PM_{2.5}$ mass concentrations are measured using gravimetric-based techniques (e.g. filters or TEOM) for regulatory purposes. Consequently, we have focused on developing a simple correction factor using

TEOM data as a reference. To explore if this was viable we plotted OPC/TEOM ratio for all sites where this was available (Bham Tyburn, Delhi, Beijing) on one plot (Fig. 8). Note we used the two-stage correction for the OPC-N2 measurements in Delhi as described in Section 3.3.1 for Fig. 8. We did not apply this correction to the Bham Tyburn data as the RH was >60 %.



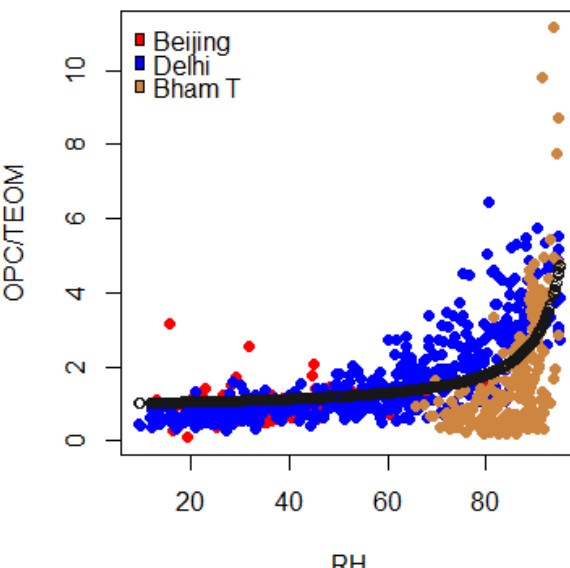

**Figure 8:** Humidogram using OPC-N2 data where there was dry reference mass (TEOM), coloured by location. The resultant κ fit (black) was generated using data from all three sites.

From Fig. 8, a κ of 0.33 (assuming a uniform particle density of 1.65 for all sites) was calculated, slightly higher

than the average of 0.3 suggested for continental regions (Pringle et al., 2010). We applied this κ of 0.33 to correct

the OPC-N2 data at all sites, as well the global average for continental regions (0.3), with the results summarised

in Table 3. Variation in the κ values generally resulted in changes to the slope, while the correlation co-efficient

remained similar. While the *in situ* derived κ resulted in a reasonable correction of the OPC-N2 relative to the

reference (±10-30 %, Table 3), using the κ from Fig. 8 (0.33) was comparable. Correcting the datasets using the

global κ of 0.33 resulted agreement with reference instruments of within 50 % at all sites, with the Beijing, Delhi

and some of the Bham Tyburn corrected OPC-N2 being within 20 % of reference (Table 3). The only site with a

notably poorer agreement using the global compared to the *in situ* κ was Bham BAQS, and this may be because

the locally derived κ for the Bham BAQS measurements (0.1) was different compared to the other sites. This

notwithstanding we do note that using a κ of 0.33 resulted in a significant improvement in accuracy compared to

the uncorrected OPC-N2 derived particle mass concentrations at Bham BAQS (slope of 3.5, Table 2). Overall,

when considering the most appropriate correction or κ the results from Table 3 suggest that a locally derived κ,

based on an *in situ* calibration with reference instrumentation, is preferable. However, the global κ derived using

data from the three urban background locations in this study (0.33, Fig. 8) gave comparable results to the *in situ*

derived correction (Table 3). Therefore, it suggests that using this κ value or suitable literature values for urban

background sites may be acceptable should there be no reference instruments available for calibration.



**Table 3:** Comparison of performance local and the global correction factors for correcting OPC-N2 using the Crilley et al. (2018) method, shown as a slope relative to the reference instrument, with $r^2$ values given in parentheses. Intercepts were not constrained to zero.

| SITE | IN SITU DERIVED CORRECTION | GLOBAL ($\kappa = 0.33$) | CONTINENTAL AVERAGE ($\kappa = 0.3$) |
|---|---|---|---|
| BHAM TYBURN | 0.98-1.33 (0.82-0.85) | 1.1-1.5 (0.82-0.85) | 1.2-1.6 (0.82-0.85) |
| DELHI | 1.1 (0.60) | 0.80 (0.61) | 0.84 (0.61) |
| BHAM BAQS | 0.96 (0.42) | 0.54 (0.45) | 0.57 (0.45) |
| BEIJING | 1.35 (0.87) | 0.85 (0.85) | 0.87 (0.85) |

## 4.0 Conclusions

Recent work has demonstrated that aerosol hygroscopocity is likely the key parameter to consider when correcting particle mass concentration derived by a low-cost OPC, particularly at high ambient RH. Consequently, correction factors have been developed that apply κ-Köhler theory to correct for the influence of water uptake by hygroscopic aerosols. In the current work, we explored the performance of this correction factor using datasets with reference instruments and a low-cost OPC (OPC-N2) co-located in environments that had differing aerosol composition, particle load and ambient RH. We observed evidence that the enhanced high concentrations reported by the OPC-N2 relative to reference instrumentation during periods of high RH was related to the amount of hygroscopic aerosols (sulphate and nitrate) and RH, as expected if the bulk aerosol hygroscopicity was driving this response. This was most clearly observed during measurements in volcanic plumes in Nicaragua, where the observed humidogram closely resembled the calculated pure sulphuric acid humidogram. This agreement between model and measurements strongly points to particle hygroscopic growth driving the high particle mass concentrations observed by the OPC-N2 during times of high ambient RH.

The results indicate that the particle mass concentration measurements reported by low-cost OPC during periods of high RH (>60 %) need to be corrected for aerosol hygroscopic growth. We employed the correction factor method outlined in Crilley et al. (2018) to account for this and observed corrected OPC-N2 $PM_{2.5}$ mass concentrations to be within 33 % of reference at all sites. The choice of applied κ was found to be critical. The results from the current work indicate that an *in situ* derived κ (using suitable reference instrumentation) leads to the most accurate correction relative to co-located reference instruments. The *in situ* derived κ would also likely be dependent on the time of year if there were any local seasonality to the bulk aerosol composition, and this would need to be considered when determining appropriate calibration procedures.

An average κ of 0.33 was calculated using measurements from three urban locations around the globe (Beijing, Birmingham, and Delhi). Applying this global κ in the correction factor notably improved reported OPC-N2 $PM_{2.5}$ mass concentrations relative to uncorrected, to be within 50 % of reference measurements at all sites. Therefore, for areas where suitable reference instrumentation for developing a local correction factor is lacking, using a literature κ value can result in a reasonable correction. For locations with low levels of hygroscopic aerosols and RH (such as Nairobi), a simple calibration against gravimetric measurements (using suitable reference instrumentation) would likely be sufficient. Whilst this study generated correction factors specific for the Alphasense OPC-N2 sensor, the calibration methodology developed is likely amenable to other low cost PM sensors.





## Data Availability

Data available on request.

## Author Contributions

LRC and FDP conceived the study. LRC, AS, LJK and FDP performed the data analysis. LRC, LJK, MDS, MSA, SY, MF, PF, YS, SG, JA, SA, RW, EI, DN, MG and FDP contributed to data investigation and curation. LRC wrote the original draft, with all other co-authors contributing to review and editing.

## Acknowledgments

We acknowledge the support from Zifa Wang and Jie Li from IAP for hosting the APHH-Beijing campaign at IAP. We thank Zongbo Shi, Di Liu, Roy Harrison and Tuan Vu from the University of Birmingham, Liangfang Wei, Hong Ren, Qiaorong Xie, Wanyu Zhao, Linjie Li, Ping Li, Shengjie Hou, Qingqing Wang from IAP, Rachel Dunmore and James Lee from the University of York, Kebin He and Xiaoting Cheng from Tsinghua University, and James Allan and Hugh Coe from the University of Manchester for providing logistic and scientific support for the field campaigns. The APHH-Beijing project was funded by the UK Natural Environment Research Council (NERC), Medical Research Council and Natural Science Foundation of China under the framework of Newton Innovation Fund (NE/N007190/1 and NE/N007077/1).

We also gratefully acknowledge the support from Prof. Mukesh Khare, Dr. Isha Kanna, Saif Khan, Rulan Verma and Prof. Gazala Habib from IIT Delhi for hosting and facilitating the measurements at IIT Delhi. LRC, LJK, MSA and WJB acknowledge funding for the ASAP-Delhi as part of APHH-India project by the UK Natural Environment Research Council (NERC), Indian Ministry of Earth Sciences (MoES) and Department for Biotechnology (DBT) (NE/P016499/1).

The work in Kenya was funded via an EPSRC grant (Global Challenges Research Fund IS2016), the Royal Society and Royal Society of Chemistry International Exchanges Award (IE170267), and DFID via the East African Research Fund (EARF) grant "A Systems Approach to Air Pollution (ASAP) East Africa". Scientific research support by International Science Programme in Sweden to the Institute of Nuclear Science & Technology in the University of Nairobi was appreciated.

The work in Nicaragua was funded by GCRF UNRESP project, grant numbers NE/P015271/1 and NE/R009465/1.

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
