# Peer review of "Effect of aerosol composition on the performance of low-cost optical particle counter correction factors"

_Atmospheric Measurement Techniques, 2019_

## Referee Comment (RC1) · Anonymous Referee #1 · 6 Nov 2019

General Comments:

This manuscript suggests a correction, (previously determined and published elsewhere), may be applied to commercial low-cost optical particle counter measurements that enable a direct comparison to PM2.5 measurements at high relative humilities in the field. Data is presented from an alphasense OPC-N2 and a TEOM-FDMS at five field locations. Whereas the OPC-N2 measures ambient (wet) aerosol size and number, which assumes a uniform particle density and reports the wet aerosol mass concentration, the TEOM-FDMS measures dry aerosol mass concentration.

The authors argue that hydroscopicity of the aerosol can be calculated from the dif-

ference in wet and dry particle mass using the OPC-N2 measurement for wet particle mass and the TEOM-FDMS measurement for dry particle mass measurement. It is difficult to accept the authors' conclusions without a clearer outline of several key assumptions.

First, the argument hinges on the idea that the OPC-N2 and the TEOM-FDMS instruments report the same aerosol size distribution in a controlled setting with low relative humidity; however this assumption is neither explicitly stated nor validated. Although the authors argue that the use of a dryer in front of the OPC-N2 measurement would render the low-cost OPC more expensive, such a lab-based comparison would substantially bolster their argument. It is disconcerting that the OPC-N2 and reference measurements are correlated but do not agree (1:1) at low humidity (<60%; e.g. Fig. 3). The reason for this disagreement is not clearly articulated. In addition, the authors do not discuss the fact that field measurements from different times of day and days likely have different compositions, which leads to large scatter in observed PM2.5 vs. relative humidity (e.g. Fig. 1 and Fig. 2) based on wind velocity and source. Finally, assumptions of spherical particles and refractive index required to calculate particle size from OPC light scattering measurements should also be discussed and are currently not mentioned. I think that this paper requires major revisions.

Specific Comments:

Table 1) Show range and mean +/- SD

Table 2) should have the same format as Table 1, with the addition of ratios to PM2.5

Figure 1 and Figure 2. Reasons for high variability in measured PM2.5 with RH should be mentioned.

Figure 3. Show 1-1 line and make the aspect ratio of figures 1:1.

Figure 4. Should this data be presented considering suspected decline in OPC-N2 performance? If so, how much of this data should be presented and how will that be

determined?

---

## Referee Comment (RC2) · Don Collins (Referee) · 19 Dec 2019

This manuscript describes the use of data collected with a low cost optical particle counter together with reference grade PM instruments to assess overall accuracy and, especially, the sensitivity to relative humidity. The manuscript is reasonably well written and understandable, though some minor editing would be required prior to publication. There is currently considerable interest in the use and performance of low cost air quality sensors and also considerable need for establishment of best practices for operation and data analysis. As noted in the manuscript, though only an Alphasense OPC-N2 was used for this analysis, the findings can at least qualitatively be extended

to the array of similar low cost PM sensors that rely on particle light scattering for measurement. Perhaps simply because of the advantage of hindsight, there are changes in techniques and instrumentation that could have provided a more easily interpreted dataset. The use of a mixture of reference instruments and the rather narrow range in humidity encountered at each of the study sites somewhat limits confidence and extension of the results. But despite the limitations of the dataset, the results would still be valuable to others using these or similar low-cost sensors and the manuscript should be publishable after the concerns identified below are addressed.

Section 2.2: There needs to be some discussion of the relationship between the reported ambient relative humidity and that in the sensor. Were the OPC-N2's inside some sort of enclosure? And if so, how was its temperature related to that outside? Would solar heating impact the temperature during the daytime? And if the sensor and outdoor temperatures are not always the same, could that help explain the large spread in derived PM2.5 at high RH mentioned on page 7, line 3?

Section 2.2: Details of the sites and reference instruments at each should be provided in a table. As is, the descriptions are structured differently enough that it is difficult for a reader to appreciate the similarities and contrasts among the sites.

Section 3.1: The relationship between RH and composition for both Delhi and Beijing should be discussed and possibly graphed. The assertions that "there is clear influence of RH on the measurements performed in Delhi..." and that the "stepwise increase in the derived particle mass between a RH of 40-50% RH may point to deliquescence..." in Beijing should be made only after quantitatively describing any role of RH-dependent variation in composition (due to connections with things such as wind patterns and photochemical production).

Figure 1: It would help to include curves representing the mean or median PM2.5 vs. RH

Figure 2: There is a brief mention in the text that the OPC-N2 tends to underestimate

PM2.5 at low RH, which partially explains the «1 OPC/TEOM ratios in Beijing (it is hard to say for the Birmingham sites because of the wide y-axis range used). But I'd like to know whether the authors have an explanation for the apparently strong dependence of the ratio on RH for RH < 40% where hygroscopic growth is probably not significant enough to be responsible. Could this be related to a confounding relationship between RH and composition?

Figure 2: Clarify in the caption that the k fits are shown in color.

Page 11, line 14: It is not necessarily true that any enhancement at RH < 79% is not due to ammonium sulfate because the particles may be in a metastable state.

Figure 3b: The sulfate content should be presented as a fraction and not as an absolute concentration. It is not surprising that sulfate mass concentration increases with increasing total (TEOM) concentration.

But that is unimportant for the OPC-N2 vs. TEOM comparison and only the fractional contribution matters.

Section 3.3: The manuscript suggests that the OPC-N2-based PM2.5 is higher than the reference and that using the measured density of 1.28 rather than the assumed density of 1.65 would not explain the discrepancy. Why not? It seems this change would improve agreement, whereas the manuscript seems to suggest it would not.

Page 13, line 9: What does "lose structure" mean here? The amplitudes of the daily peaks are lower than earlier in the measurement period, but why should that indicate there is a problem? This is especially important because it is the basis for the suggestion that the lifetime of the sensor in Delhi is only a month. And furthermore, even if the sensitivity of the OPC-N2 decreased due to dirty optics, the authors should not attempt to extrapolate to all low cost sensors as they do here.

Page 14, line 11: Does "...though within a large range" refer to 0.1 – 0.12? If so, I don't think of that as a large range. And if not, reword this so it is clearer.

Section 3.4, first paragraph: Rather than relying on trajectories, it seems statistical tools or machine learning could provide at least some improvement in correction accuracy using things like weather observations, day of the week, time of day. . .

Page 16: Use a word other than global, which implies something based on more than three urban background sites during only relatively short periods of the year.

---

## Author Comment (AC1) · 27 Jan 2020

We thank the reviewers for their comments on our manuscript. Please find our detailed responses to the comments from both reviewers below in red.

**Response to Reviewer 1**

General Comments: This manuscript suggests a correction, (previously determined and published elsewhere), may be applied to commercial low-cost optical particle counter measurements that enable a direct comparison to PM2.5 measurements at high relative humilities in the field.

It is true that both Crilley et al. (2018) and Di Antonio et al. (2018) have previously provided correction methodologies, but these were for specific locations with specific meteorology. It is important to note that a universal correction is not possible for individual particles due to differences in composition (hence hygroscopicity), size and shape. Instead, if it is possible to correct for an ensemble of aerosols, using average hygroscopicity, this would make it significantly easier to correct low-cost optical particle counters without having to employ detailed calibrations on the particle size distribution. Therefore, the main aim of this paper is to investigate whether a universal correction method can be reasonably applied to reported particle mass concentrations by low-cost OPC across a wide variety of locations. By achieving this aim (with associated errors), we believe this paper of significant use to the AMT readership. To clarify the aims, we have added the following sentence to the end of the introduction:

**"The aim of this paper was to investigate whether a universal correction method can be reasonably applied to reported particle mass concentrations by low-cost OPC across a wide variety of locations."**

Data is presented from an alphasense OPC-N2 and a TEOM-FDMS at five field locations. Whereas the OPC-N2 measures ambient (wet) aerosol size and number, which assumes a uniform particle density and reports the wet aerosol mass concentration, the TEOM-FDMS measures dry aerosol mass concentration The authors argue that hydroscopicity of the aerosol can be calculated from the difference in wet and dry particle mass using the OPC-N2 measurement for wet particle mass and the TEOM-FDMS measurement for dry particle mass measurement. It is difficult to accept the authors' conclusions without a clearer outline of several key assumptions.

First, the argument hinges on the idea that the OPC-N2 and the TEOM-FDMS instruments report the same aerosol size distribution in a controlled setting with low relative humidity; however this assumption is neither explicitly stated nor validated.

Previous work has shown that a calibration for the reported OPC-N2 particle mass concentrations (as well as other low-cost OPC) is often needed with respect to reference instrumentation (e.g. TEOM-FDMS). This has been investigated in depth both in the field (see e.g. Crilley et al, Di Antonio et al, Pope et al. and in the lab (e.g. Sousan et al. 2016).

As mentioned by the reviewer, there are several assumptions made in relating the OPC and TEOM measurements to each other. These assumptions include particle sphericity, density and size distribution. Firstly, we are using the reported particle mass measurements by the OPC-N2 throughout. In the conversion from particle number distribution, it is assumed that the particles are all spherical with a uniform density across the particle size distribution measured by the OPC-N2. In addition, it is also assumed that the particle mass below the OPC-N2 particle size cut-off (300 nm) is inconsequential. The key assumption made is that both the OPC-N2 and TEOM are responding to the same dry aerosol mass for particles below the cut-off for the TEOM-FDMS (PM2.5). The work by Di Antonio et al (2018)

demonstrated that with correction, the OPC-N2 can report the same dry aerosol size distribution as reference instrument.

To clarify these assumptions, the following text has been added to section 2.3:

"To calculate the particle mass concentration from measured particle number size distribution, spherical particles of a uniform density and shape are assumed by the OPC, which is not strictly true for airborne particles in an urban atmosphere but is considered a standard approximation. For full details see Crilley et al. (2018). The OPC-N2 assumes the ambient particle density to be 1.65 g cm-3 across all size bins to derive the particle mass concentrations from the measured particles number concentrations (Crilley et al., 2018), therefore we have used this density for the dry particles (pp) in Eq. (2). We assume that the particle density is uniform across the particle size distribution measured by the OPC-N2. Furthermore, we assume that both the OPC-N2 and reference instrument are responding to same dry aerosol mass for the all particles below the size cut-off on the reference instrument. We also note that we assume both the OPC-N2 and reference instrument responses are linear over the range of measured concentrations at each site."

Although the authors argue that the use of a dryer in front of the OPC-N2 measurement would render the low-cost OPC more expensive, such a lab-based comparison would substantially bolster their argument.

It is true that lab-based study to exploring the response of the OPC-N2 fitted with a dryer before the inlet would be useful, but this is outside the scope of the current work. We aimed to explore the universal applicability of the correction factor by using a number of field datasets. We do note that previous work by Sousan et al (2016) investigated the response of the OPC-N2 under low RH in a controlled laboratory setting and found good agreement in reported mass concentrations with GRIMM PAS-1.108 ( $r_2 > 0.97$ ) but the slopes varied depending on the aerosol composition. This would suggest that drying the aerosol before measurement by the OPC-N2 could improve the accuracy, but it would not negate the need for in-situ calibration.

It is disconcerting that the OPC-N2 and reference measurements are correlated but do not agree (1:1) at low humidity (e.g. <60%; Fig. 3). The reason for this disagreement is not clearly articulated.

This is likely due to the previously stated differences in density, composition, size and shape of individual particles. Some particle-phase chemical species are more likely to be found in high concentrations at different times of the day due to changes in sources and sinks. This is demonstrated by recent high temporal aerosol composition measurements using an aerosol mass spectrometer (AMS) in urban locations where a distinct diurnal trend in bulk aerosol composition (e.g. ammonium, sulphates, nitrates, organics etc) was observed, see e.g. Young et al 2014.

In addition, the authors do not discuss the fact that field measurements from different times of day and days likely have different compositions, which leads to large scatter in observed PM2.5 vs. relative humidity (e.g. Fig. 1 and Fig. 2) based on wind velocity and source.

The reviewer makes a good point. Therefore, we have added the following sentence to the end of section 3.1 to clarify this.

"The scatter in OPC/TEOM observed in Fig 2 as function of RH was likely due to temporal variability in aerosol composition due to changing sources and sinks (both local and regional)."

Finally, assumptions of spherical particles and refractive index required to calculate particle size from OPC light scattering measurements should also be discussed and are currently not mentioned.

We have included text outlining these assumptions in Section 2.3, please see our earlier response.

I think that this paper requires major revisions.

Specific Comments:

Table 1) Show range and mean +/- SD

We have made these changes to Table 1, as shown below

**Table 1:** Summary of measurement datasets. Reported OPC-N2 PM2.5 mass concentrations are uncorrected. For the Nicaragua measurements there was no co-located reference instrumentation. Only one 24 hr average gravimetric PM2.5 concentration was available for Nairobi, presented with stated measurement uncertainty.

| Site       | Date         |              | RH      | OPC-N2                                  | Reference                               |  |
|------------|--------------|--------------|---------|-----------------------------------------|-----------------------------------------|--|
|            |              |              | (%)     | PM 2.5 (μg m -3 ) | PM 2.5 (μg m -3 ) |  |
| Birmingham | Oct 2016-    | Average ± SD | 89±10   | 38 ± 54                                 | 11 ± 8                                  |  |
|            | Feb 2017     | Range        | 44 - 99 | 0.3 - 566                               | 0.5 - 63                                |  |
| Beijing    | Dec 2016     | Average ± SD | 36±15   | 74±82                                   | 75±66                                   |  |
|            |              | Range        | 13 - 81 | 3 - 274                                 | 2.7 - 208                               |  |
| Delhi      | Jan-Feb 2018 | Average ± SD | 59±25   | 218±228                                 | 168±76                                  |  |
|            |              | Range        | 9 - 100 | 12 - 1113                               | 50 - 478                                |  |
| Nairobi    | Feb-Mar 2017 | Average ± SD | 51±18   | 32±16                                   | 27 646 9                                |  |
|            |              | Range        | 16 - 89 | 4 - 135                                 | 27.010.0                                |  |
| Nicaragua  | Feb-Dec 2017 | Average ± SD | 77±11   | 21±55                                   | NIA                                     |  |
|            |              | Range        | 39 - 91 | 0.5 - 742                               | INA                                     |  |

Table 2) should have the same format as Table 1, with the addition of ratios to PM2.5

We do not understand what the reviewer is asking for here, as Table 2 reports slopes and consequently have not made any changes to Table 2.

Figure 1 and Figure 2. Reasons for high variability in measured PM2.5 with RH should be mentioned.

Please see our response to earlier comment.

Figure 3. Show 1-1 line and make the aspect ratio of figures 1:1.

These changes have been made.

Figure 4. Should this data be presented considering suspected decline in OPC-N2 performance? If so, how much of this data should be presented and how will that be determined?

We have re-evaluated the data and we now do not feel there is enough evidence to suggest a decline in performance. As pointed out by reviewer 2, just because there appears to be change in amplitude in the reported PM2.5 mass concentration in the last 2 days (Fig 4), this does not mean that there is a decline in performance. The time series in Fig 4 is not long enough to ascertain this, and there we have removed this discussion on suspected declining performance from the text and will show the whole dataset. The paragraph now reads:

"During the measurements in Delhi, the OPC-N2 typically over-reported the PM2.5 mass concentrations relative to the reference (Fig. 6A). The OPC-N2 assumes a uniform particle density of 1.65 g cm-3 in the particle counts to mass conversion, and this density may be inappropriate for Delhi aerosol during winter. Previous measurements of aerosol density during winter in Delhi at midday were on average 1.28±0.12 g cm-3 (Sarangi et al., 2016), lower than applied by the OPC-N2. Generally, the OPC/TEOM ratio was below 1 (Fig 2A) and so this would not explain this discrepancy. From Fig. 6A, higher PM2.5 mass concentrations relative to the reference were reported by the OPC-N2 towards the start of the measurement period (Fig. 6A), generally during the times when the RH was high (e.g. 25-29th Jan, Fig. 6B). We also note that the reported concentrations by the OPC-N2 towards the end of the measurements in Delhi (11th Feb onwards, Fig. 6A), were in better agreement with reference. The cause of this change in performance is unclear but could reflect lower RH or changes in aerosol composition"

**Response to Reviewer 2 (Don Collins)**

This manuscript describes the use of data collected with a low cost optical particle counter together with reference grade PM instruments to assess overall accuracy and, especially, the sensitivity to relative humidity. The manuscript is reasonably well written and understandable, though some minor editing would be required prior to publication. There is currently considerable interest in the use and performance of low cost air quality sensors and also considerable need for establishment of best practices for operation and data analysis. As noted in the manuscript, though only an Alphasense OPC-N2 was used for this analysis, the findings can at least qualitatively be extended to the array of similar low cost PM sensors that rely on particle light scattering for measurement. Perhaps simply because of the advantage of hindsight, there are changes in techniques and instrumentation that could have provided a more easily interpreted dataset. The use of a mixture of reference instruments and the rather narrow range in humidity encountered at each of the study sites somewhat limits confidence and extension of the results. But despite the limitations of the dataset, the results would still be valuable to others using these or similar low-cost sensors and the manuscript should be publishable after the concerns identified below are addressed.

**We thank the reviewer for their positive comments.**

We agree that examining different low-cost OPC would have been useful but were limited by available instrumentation during the respective field campaigns. We would like to note that we primarily used the TEOM-FDMS as reference instrument at the sites of primary interest in the manuscript (Beijing, Delhi and Birmingham Tyburn).

Section 2.2: There needs to be some discussion of the relationship between the reported ambient relative humidity and that in the sensor. Were the OPC-N2's inside some sort of enclosure? And if so,

how was its temperature related to that outside? Would solar heating impact the temperature during the daytime? And if the sensor and outdoor temperatures are not always the same, could that help explain the large spread in derived PM2.5 at high RH mentioned on page 7, line 3?

Yes, the OPC-N2 were housed inside a small enclosure that was placed outside the main laboratory at all sites. As such we expect the internal temperature of OPC-N2 to be similar to ambient, but we do not have any measurements to support this.

Section 2.2: Details of the sites and reference instruments at each should be provided in a table. As is, the descriptions are structured differently enough that it is difficult for a reader to appreciate the similarities and contrasts among the sites.

We have added the following table to section 2 as a summary of the different sites in the manuscript.

Table 1: Summary of the measurement sites. Full details available in the text. N/A signifies not available. Custom housing for the OPC-N2 as per description in text.

| Location        | Site        | OPC-N2  | Reference   | Aerosol     |
|-----------------|-------------|---------|-------------|-------------|
|                 | Description | housing | instrument  | composition |
|                 |             |         |             | instrument  |
| Birmingham      | Urban       | Custom  | GRIMM       | N/A         |
| BAQS, UK        | background  |         |             |             |
| Birmingham      | Urban       | Custom  | TEOM-FDMS   | N/A         |
| Tyburn, UK      | background  |         |             |             |
| Beijing, China  | Urban       | Custom  | TEOM-FDMS   | AMS         |
|                 | background  |         |             |             |
| Delhi, India    | Urban       | Custom  | TEOM-FDMS   | ACSM        |
|                 | background  |         |             |             |
| Nairobi, Kenya  | Urban       | Custom  | Gravimetric | N/A         |
|                 | background  |         |             |             |
| Masaya volcano, | Volcano     | AQMESH  | N/A         | N/A         |
| Nicaragua       |             |         |             |             |

Section 3.1: The relationship between RH and composition for both Delhi and Beijing should be discussed and possibly graphed. The assertions that "there is clear influence of RH on the measurements performed in Delhi" and that the "stepwise increase in the derived particle mass between a RH of 40-50% RH may point to deliquescence" in Beijing should be made only after quantitatively describing any role of RH-dependent variation in composition (due to connections with things such as wind patterns and photochemical production).

Later in the manuscript, we state that explore the relationship RH, aerosol composition and OPC-N2 reported mass concentrations in section 3.2 for Delhi and Beijing, as shown in Figs 3 and 4.

Figure 1: It would help to include curves representing the mean or median PM2.5 vs RH

We think that including curves of the mean PM2.5 vs RH could be misleading as we want to highlight the range of concentrations measured as function of RH. We have therefore added the mean reported PM2.5 mass concentration binned by RH to figures in supporting information, as shown below.

---

## Author Comment (AC2) · 29 Jan 2020

We thank the reviewers for their comments on our manuscript. Please find our detailed responses to the comments from both reviewers below in red.

**Response to Reviewer 1**

General Comments: This manuscript suggests a correction, (previously determined and published elsewhere), may be applied to commercial low-cost optical particle counter measurements that enable a direct comparison to PM2.5 measurements at high relative humilities in the field.

It is true that both Crilley et al. (2018) and Di Antonio et al. (2018) have previously provided correction methodologies, but these were for specific locations with specific meteorology. It is important to note that a universal correction is not possible for individual particles due to differences in composition (hence hygroscopicity), size and shape. Instead, if it is possible to correct for an ensemble of aerosols, using average hygroscopicity, this would make it significantly easier to correct low-cost optical particle counters without having to employ detailed calibrations on the particle size distribution. Therefore, the main aim of this paper is to investigate whether a universal correction method can be reasonably applied to reported particle mass concentrations by low-cost OPC across a wide variety of locations. By achieving this aim (with associated errors), we believe this paper of significant use to the AMT readership. To clarify the aims, we have added the following sentence to the end of the introduction:

*"The aim of this paper was to investigate whether a universal correction method can be reasonably applied to reported particle mass concentrations by low-cost OPC across a wide variety of locations."*

Data is presented from an alphasense OPC-N2 and a TEOM-FDMS at five field locations. Whereas the OPC-N2 measures ambient (wet) aerosol size and number, which assumes a uniform particle density and reports the wet aerosol mass concentration, the TEOM-FDMS measures dry aerosol mass concentration The authors argue that hydroscopicity of the aerosol can be calculated from the difference in wet and dry particle mass using the OPC-N2 measurement for wet particle mass and the TEOM-FDMS measurement for dry particle mass measurement. It is difficult to accept the authors' conclusions without a clearer outline of several key assumptions.

First, the argument hinges on the idea that the OPC-N2 and the TEOM-FDMS instruments report the same aerosol size distribution in a controlled setting with low relative humidity; however this assumption is neither explicitly stated nor validated.

Previous work has shown that a calibration for the reported OPC-N2 particle mass concentrations (as well as other low-cost OPC) is often needed with respect to reference instrumentation (e.g. TEOM-FDMS). This has been investigated in depth both in the field (see e.g. Crilley et al, Di Antonio et al, Pope et al. and in the lab (e.g. Sousan et al. 2016).

As mentioned by the reviewer, there are several assumptions made in relating the OPC and TEOM measurements to each other. These assumptions include particle sphericity, density and size distribution. Firstly, we are using the reported particle mass measurements by the OPC-N2 throughout. In the conversion from particle number distribution, it is assumed that the particles are all spherical with a uniform density across the particle size distribution measured by the OPC-N2. In addition, it is also assumed that the particle mass below the OPC-N2 particle size cut-off (300 nm) is inconsequential. The key assumption made is that both the OPC-N2 and TEOM are responding to the same dry aerosol mass for particles below the cut-off for the TEOM-FDMS (PM2.5). The work by Di Antonio et al (2018)

demonstrated that with correction, the OPC-N2 can report the same dry aerosol size distribution as reference instrument.

To clarify these assumptions, the following text has been added to section 2.3:

*"To calculate the particle mass concentration from measured particle number size distribution, spherical particles of a uniform density and shape are assumed by the OPC, which is not strictly true for airborne particles in an urban atmosphere but is considered a standard approximation. For full details see Crilley et al. (2018). The OPC-N2 assumes the ambient particle density to be 1.65 g cm-3 across all size bins to derive the particle mass concentrations from the measured particle number concentrations (Crilley et al., 2018), therefore we have used this density for the dry particles ($\rho p$) in Eq. (2). We assume that the particle density is uniform across the particle size distribution measured by the OPC-N2. Furthermore, we assume that both the OPC-N2 and reference instrument are responding to same dry aerosol mass for the all particles below the size cut-off on the reference instrument. We also note that we assume both the OPC-N2 and reference instrument responses are linear over the range of measured concentrations at each site."*

Although the authors argue that the use of a dryer in front of the OPC-N2 measurement would render the low-cost OPC more expensive, such a lab-based comparison would substantially bolster their argument.

It is true that lab-based study to exploring the response of the OPC-N2 fitted with a dryer before the inlet would be useful, but this is outside the scope of the current work. We aimed to explore the universal applicability of the correction factor by using a number of field datasets. We do note that previous work by Sousan et al (2016) investigated the response of the OPC-N2 under low RH in a controlled laboratory setting and found good agreement in reported mass concentrations with GRIMM PAS-1.108 ($r2 > 0.97$) but the slopes varied depending on the aerosol composition. This would suggest that drying the aerosol before measurement by the OPC-N2 could improve the accuracy, but it would not negate the need for in-situ calibration.

It is disconcerting that the OPC-N2 and reference measurements are correlated but do not agree (1:1) at low humidity (e.g. <60%; Fig. 3). The reason for this disagreement is not clearly articulated.

This is likely due to the previously stated differences in density, composition, size and shape of individual particles. Some particle-phase chemical species are more likely to be found in high concentrations at different times of the day due to changes in sources and sinks. This is demonstrated by recent high temporal aerosol composition measurements using an aerosol mass spectrometer (AMS) in urban locations where a distinct diurnal trend in bulk aerosol composition (e.g. ammonium, sulphates, nitrates, organics etc) was observed, see e.g. Young et al 2014.

In addition, the authors do not discuss the fact that field measurements from different times of day and days likely have different compositions, which leads to large scatter in observed PM2.5 vs. relative humidity (e.g. Fig. 1 and Fig. 2) based on wind velocity and source.

The reviewer makes a good point. Therefore, we have added the following sentence to the end of section 3.1 to clarify this.

*"The scatter in OPC/TEOM observed in Fig 2 as function of RH was likely due to temporal variability in aerosol composition due to changing sources and sinks (both local and regional)."*

Finally, assumptions of spherical particles and refractive index required to calculate particle size from OPC light scattering measurements should also be discussed and are currently not mentioned.

We have included text outlining these assumptions in Section 2.3, please see our earlier response.

I think that this paper requires major revisions.

Specific Comments:

Table 1) Show range and mean +/- SD

We have made these changes to Table 1, as shown below

**Table 1:** Summary of measurement datasets. Reported OPC-N2 $PM_{2.5}$ mass concentrations are uncorrected. For the Nicaragua measurements there was no co-located reference instrumentation. Only one 24 hr average gravimetric $PM_{2.5}$ concentration was available for Nairobi, presented with stated measurement uncertainty.

| Site | Date | | RH (%) | OPC-N2 $PM_{2.5}$ ($\mu g\ m^{-3}$) | Reference $PM_{2.5}$ ($\mu g\ m^{-3}$) |
|---|---|---|---|---|---|
| Birmingham | Oct 2016- | Average ± SD | 89±10 | 38 ± 54 | 11 ± 8 |
| | Feb 2017 | Range | 44 - 99 | 0.3 - 566 | 0.5 - 63 |
| Beijing | Dec 2016 | Average ± SD | 36±15 | 74±82 | 75±66 |
| | | Range | 13 - 81 | 3 - 274 | 2.7 - 208 |
| Delhi | Jan-Feb 2018 | Average ± SD | 59±25 | 218±228 | 168±76 |
| | | Range | 9 - 100 | 12 - 1113 | 50 - 478 |
| Nairobi | Feb-Mar 2017 | Average ± SD | 51±18 | 32±16 | 27.6±6.8 |
| | | Range | 16 - 89 | 4 - 135 | |
| Nicaragua | Feb-Dec 2017 | Average ± SD | 77±11 | 21±55 | NA |
| | | Range | 39 - 91 | 0.5 - 742 | |

Table 2) should have the same format as Table 1, with the addition of ratios to PM2.5

We do not understand what the reviewer is asking for here, as Table 2 reports slopes and consequently have not made any changes to Table 2.

Figure 1 and Figure 2. Reasons for high variability in measured PM2.5 with RH should be mentioned.

Please see our response to earlier comment.

Figure 3. Show 1-1 line and make the aspect ratio of figures 1:1.

These changes have been made.

Figure 4. Should this data be presented considering suspected decline in OPC-N2 performance? If so, how much of this data should be presented and how will that be determined?

We have re-evaluated the data and we now do not feel there is enough evidence to suggest a decline in performance. As pointed out by reviewer 2, just because there appears to be change in amplitude in the reported PM2.5 mass concentration in the last 2 days (Fig 4), this does not mean that there is a decline in performance. The time series in Fig 4 is not long enough to ascertain this, and there we have removed this discussion on suspected declining performance from the text and will show the whole dataset. The paragraph now reads:

*"During the measurements in Delhi, the OPC-N2 typically over-reported the PM2.5 mass concentrations relative to the reference (Fig. 6A). The OPC-N2 assumes a uniform particle density of 1.65 g cm-3 in the particle counts to mass conversion, and this density may be inappropriate for Delhi aerosol during winter. Previous measurements of aerosol density during winter in Delhi at midday were on average 1.28±0.12 g cm-3 (Sarangi et al., 2016), lower than applied by the OPC-N2. Generally, the OPC/TEOM ratio was below 1 (Fig 2A) and so this would not explain this discrepancy. From Fig. 6A, higher PM2.5 mass concentrations relative to the reference were reported by the OPC-N2 towards the start of the measurement period (Fig. 6A), generally during the times when the RH was high (e.g. 25-29th Jan, Fig. 6B). We also note that the reported concentrations by the OPC-N2 towards the end of the measurements in Delhi (11th Feb onwards, Fig. 6A), were in better agreement with reference. The cause of this change in performance is unclear but could reflect lower RH or changes in aerosol composition"*

**Response to Reviewer 2 (Don Collins)**

This manuscript describes the use of data collected with a low cost optical particle counter together with reference grade PM instruments to assess overall accuracy and, especially, the sensitivity to relative humidity. The manuscript is reasonably well written and understandable, though some minor editing would be required prior to publication. There is currently considerable interest in the use and performance of low cost air quality sensors and also considerable need for establishment of best practices for operation and data analysis. As noted in the manuscript, though only an Alphasense OPC-N2 was used for this analysis, the findings can at least qualitatively be extended to the array of similar low cost PM sensors that rely on particle light scattering for measurement. Perhaps simply because of the advantage of hindsight, there are changes in techniques and instrumentation that could have provided a more easily interpreted dataset. The use of a mixture of reference instruments and the rather narrow range in humidity encountered at each of the study sites somewhat limits confidence and extension of the results. But despite the limitations of the dataset, the results would still be valuable to others using these or similar low-cost sensors and the manuscript should be publishable after the concerns identified below are addressed.

We thank the reviewer for their positive comments.
We agree that examining different low-cost OPC would have been useful but were limited by available instrumentation during the respective field campaigns. We would like to note that we primarily used the TEOM-FDMS as reference instrument at the sites of primary interest in the manuscript (Beijing, Delhi and Birmingham Tyburn).

Section 2.2: There needs to be some discussion of the relationship between the reported ambient relative humidity and that in the sensor. Were the OPC-N2's inside some sort of enclosure? And if so,

how was its temperature related to that outside? Would solar heating impact the temperature during the daytime? And if the sensor and outdoor temperatures are not always the same, could that help explain the large spread in derived PM2.5 at high RH mentioned on page 7, line 3?

Yes, the OPC-N2 were housed inside a small enclosure that was placed outside the main laboratory at all sites. As such we expect the internal temperature of OPC-N2 to be similar to ambient, but we do not have any measurements to support this.

Section 2.2: Details of the sites and reference instruments at each should be provided in a table. As is, the descriptions are structured differently enough that it is difficult for a reader to appreciate the similarities and contrasts among the sites.

We have added the following table to section 2 as a summary of the different sites in the manuscript.

Table 1: Summary of the measurement sites. Full details available in the text. N/A signifies not available. Custom housing for the OPC-N2 as per description in text.

| Location | Site Description | OPC-N2 housing | Reference instrument | Aerosol composition instrument |
|---|---|---|---|---|
| Birmingham BAQS, UK | Urban background | Custom | GRIMM | N/A |
| Birmingham Tyburn, UK | Urban background | Custom | TEOM-FDMS | N/A |
| Beijing, China | Urban background | Custom | TEOM-FDMS | AMS |
| Delhi, India | Urban background | Custom | TEOM-FDMS | ACSM |
| Nairobi, Kenya | Urban background | Custom | Gravimetric | N/A |
| Masaya volcano, Nicaragua | Volcano | AQMESH | N/A | N/A |

Section 3.1: The relationship between RH and composition for both Delhi and Beijing should be discussed and possibly graphed. The assertions that "there is clear influence of RH on the measurements performed in Delhi" and that the "stepwise increase in the derived particle mass between a RH of 40-50% RH may point to deliquescence" in Beijing should be made only after quantitatively describing any role of RH-dependent variation in composition (due to connections with things such as wind patterns and photochemical production).

Later in the manuscript, we state that explore the relationship RH, aerosol composition and OPC-N2 reported mass concentrations in section 3.2 for Delhi and Beijing, as shown in Figs 3 and 4.

Figure 1: It would help to include curves representing the mean or median PM2.5 vs RH

We think that including curves of the mean PM2.5 vs RH could be misleading as we want to highlight the range of concentrations measured as function of RH. We have therefore added the mean reported PM2.5 mass concentration binned by RH to figures in supporting information, as shown below.

[Figure]

[Figure]

Figure S1: Plot of reported PM2.5 mass concentration by OPC-N2 against ambient RH for the whole measurement period in Delhi (A), Nairobi (B), Beijing (C), Bham BAQS (D), Nicaragua (E), Bham Tyburn (F). The black line is the mean reported PM2.5 mass concentration binned by RH. Note the different y- and x-axis scales.

Figure 2: There is a brief mention in the text that the OPC-N2 tends to underestimate PM2.5 at low RH, which partially explains the «1 OPC/TEOM ratios in Beijing (it is hard to say for the Birmingham sites because of the wide y-axis range used). But I'd like to know whether the authors have an explanation for the apparently strong dependence of the ratio on RH for RH < 40% where hygroscopic growth is probably not significant enough to be responsible. Could this be related to a confounding relationship between RH and composition?

The reviewer asks some pertinent questions. It should be noted that aerosol with high hygroscopicity, typically those with high mass ratios of inorganic salts, can take up water significantly at RHs less than 50%, for example sulphates and nitrates.  However, we cannot rule out correlations between RH and composition that are not related to hygroscopicity. It is possible that the low RH correlates with low particle concentrations independent of hygroscopicity.

Figure 2: Clarify in the caption that the k fits are shown in color.

We have changed the caption to read:

*"Figure 2: Humidograms with corresponding κ fit (shown in colour) for Delhi (A), Beijing (B), Bham BAQS (C), Bham Tyburn (D). Note the different y-axis and x-axis scales. The two-stage correction factor described in Section 3.3.1 has not been applied for these humidograms."*

Page 11, line 14: It is not necessarily true that any enhancement at RH < 79% is not due to ammonium sulfate because the particles may be in a metastable state.

The reviewer is correct that ammonium sulphate could be in a metastable state and hence not efflorescent. We have rephrased the passage to reflect this:

*"From Figs. 3 and 4, it appears that this effect was occurring at RH above 40 %, below the deliquescence point of ammonium sulphate (79 %) indicating the ammonium sulphate component of the aerosol was in a metastable state. Aerosols with multi-component mixtures are observed to deliquesce earlier than the deliquescence points of the individual components e.g. Pope et al. (2010). It is noted that the nitrate component of the aerosols have a smoother continual take up of water with respect to RH (Gibson et al., 2006;Hu et al., 2010)."*

Figure 3b: The sulfate content should be presented as a fraction and not as an absolute concentration. It is not surprising that sulfate mass concentration increases with increasing total (TEOM) concentration. But that is unimportant for the OPC-N2 vs. TEOM comparison and only the fractional contribution matters.

We note that the sulphate concentrations were obtained using an aerosol mass spectrometer, which measures PM1, while the TEOM measures PM2.5. therefore, this fraction is only an estimate as a substantial fraction of sulphate (PM1-2.5) missing.
But the reviewer has made a valid point and we have made the change to Fig 3b as shown below and updated the text to reflect this change

[Figure]

[Figure]

Figure 3: Reported OPC-N2 uncorrected PM2.5 mass concentrations against TEOM PM2.5 mass concentration measurements coloured by ambient RH (A) and fraction of sulphate to the total PM2.5 mass (B) in Beijing. The straight line indicates the linear regression fit for concentrations below 150 µg m-3. The dashed line is 1:1.

Section 3.3: The manuscript suggests that the OPC-N2-based PM2.5 is higher than the reference and that using the measured density of 1.28 rather than the assumed density of 1.65 would not explain the discrepancy. Why not? It seems this change would improve agreement, whereas the manuscript seems to suggest it would not.

While there are periods where the reported concentration by OPC-N2 was considerably higher than by the TEOM (reference) in Delhi, generally the ratio of OPC/TEOM was below 1, as shown in Figure 2A.

Therefore, using a lower particle density would not improve the agreement between the OPC and reference instrument. To clarify this point, we have changed the text to read:

*"Previous measurements of aerosol density during winter in Delhi at midday were on average 1.28±0.12 g cm-3 (Sarangi et al., 2016), which is lower than that applied by the OPC-N2 (1.65). Generally, the OPC/TEOM ratio was below 1 (Fig 2A) and so this would not fully explain this discrepancy."*

Page 13, line 9: What does "lose structure" mean here? The amplitudes of the daily peaks are lower than earlier in the measurement period, but why should that indicate there is a problem? This is especially important because it is the basis for the suggestion that the lifetime of the sensor in Delhi is only a month. And furthermore, even if the sensitivity of the OPC-N2 decreased due to dirty optics, the authors should not attempt to extrapolate to all low cost sensors as they do here.

The reviewer has made a valid point and we agree that we do not have enough evidence to suggest that there was a problem with sensor. Consequently, we have edited this paragraph to remove this discussion.

Please see our response to the last comment from Reviewer 1.

Page 14, line 11: Does "though within a large range" refer to 0.1 – 0.12? If so, I don't think of that as a large range. And if not, reword this so it is clearer.

We were referring to the large scatter OPC/TEOM in Fig 2C and not that calculated kappa values.
To clarify this point, we have changed the text to read:

*"The calculated κ value for each month in Birmingham had little variability (0.1-0.12) from October to January and this suggests that on average the bulk hygroscopicity of the aerosols was consistent, though within a large range (as indicated by the large spread in OPC/TEOM in Fig 2c)."*

Section 3.4, first paragraph: Rather than relying on trajectories, it seems statistical tools or machine learning could provide at least some improvement in correction accuracy using things like weather observations, day of the week, time of day…

We applied this method of back trajectories as a proxy for particle composition (and hence hygroscopicity) as it was successfully applied by Di Antonio et al. (2018) for their dataset, to see if could also be applied to our dataset. Statistical tools may be able to improve the correction accuracy but are outside the scope of this work.

Page 16: Use a word other than global, which implies something based on more than three urban background sites during only relatively short periods of the year.

We chose to refer to this calculated kappa value as global as it covered urban background sites across 2 continents.  The reviewer does make a valid point, and we have added the following sentence to clarify:

*"From Fig. 8, a κ of 0.33 (assuming a uniform particle density of 1.65 for all sites) was calculated, slightly higher than the average of 0.3 suggested for continental regions (Pringle et al., 2010). Consequently, we refer to the calculated κ from Fig 8 as global κ but note that it was calculated from 3 urban background*

*sites on two continents. We applied this κ (0.33) to correct the OPC-N2 data at all sites, as well the average for continental regions (0.3), with the results summarised in Table 3."*

**References**

Crilley, L.R., Shaw, M., Pound, R., Kramer, L.J., Price, R., Young, S., Lewis, A.C. and Pope, F.D., 2018. Evaluation of a low-cost optical particle counter (Alphasense OPC-N2) for ambient air monitoring. *Atmospheric Measurement Techniques*, pp.709-720.

Di Antonio, A., Popoola, O., Ouyang, B., Saffell, J. and Jones, R., 2018. Developing a relative humidity correction for low-cost sensors measuring ambient particulate matter. *Sensors*, *18*(9), p.2790.

Sousan, S., Koehler, K., Hallett, L. and Peters, T.M., 2016. Evaluation of the Alphasense optical particle counter (OPC-N2) and the Grimm portable aerosol spectrometer (PAS-1.108). *Aerosol Science and Technology*, *50*(12), pp.1352-1365.

Young, D.E., Allan, J.D., Williams, P.I., Green, D.C., Flynn, M.J., Harrison, R.M., Yin, J., Gallagher, M.W. and Coe, H., 2015. Investigating the annual behaviour of submicron secondary inorganic and organic aerosols in London. *Atmospheric Chemistry and Physics*, *15*(11), pp.6351-6366.